# Erectile Dysfunction Treatment Using Stem Cell Delivery Patch in a Cavernous Nerve Injury Rat Model

**DOI:** 10.3390/bioengineering10060635

**Published:** 2023-05-24

**Authors:** Hyong Woo Moon, In Gul Kim, Mee Young Kim, Ae Ryang Jung, Kwideok Park, Ji Youl Lee

**Affiliations:** 1Department of Urology, Seoul St. Mary’s Hospital, College of Medicine, The Catholic University of Korea, Seoul 06591, Republic of Korea; aspasias@catholic.ac.kr (H.W.M.); exactly@nate.com (A.R.J.); 2Department of Otorhinolaryngology-Head and Neck Surgery, Seoul National University Hospital, Seoul 03080, Republic of Korea; biokim@gmail.com; 3Catholic Cancer Research Institute, College of Medicine, The Catholic University of Korea, Seoul 06591, Republic of Korea; 4Center for Biomaterials, Korea Institute of Science and Technology (KIST), Seoul 02792, Republic of Korea

**Keywords:** rectile dysfunction, stem cell delivery, prostate cancer, biofabrication

## Abstract

Erectile dysfunction (ED) is a common and feared complication of radical prostatectomy (RP) for prostate cancer. Recently, tissue engineering for post-prostatectomy ED has been attempted in which controlled interactions between cells, growth factors, and the extracellular matrix (ECM) are important for the structural integrity if nerve regeneration. In this study, we evaluated the effects of a biomechanical ECM patch on the morphology and behavior of human bone marrow-derived mesenchymal stem cells (hBMSCs) in a bilateral cavernous nerve injury (BCNI) rat model. The ECM patch, made of decellularized human fibroblast-derived ECM (hFDM) and a biocompatible polyvinyl alcohol (PVA) hydrogel, was tested with human bone marrow-derived mesenchymal stem cells (hBMSCs) on a bilateral cavernous nerve injury (BCNI) rat model. In vitro analysis showed that the hFDM/PVA + hBMSCs patches significantly increased neural development markers. In vivo experiments demonstrated that the rats treated with the hFDM/PVA patch had higher ICP/MAP ratios, higher ratios of smooth muscle to collagen, increased nNOS content, higher levels of eNOS protein expression, and higher cGMP levels compared to the BCNI group. These results indicate that the hFDM/PVA patch is effective in promoting angiogenesis, smooth muscle regeneration, and nitrergic nerve regeneration, which could contribute to improved erectile function in post-prostatectomy ED.

## 1. Introduction

Erectile dysfunction (ED) is a common and feared complication of radical prostatectomy (RP) for prostate cancer, with many patients seeing recovery of their ability to have an erection as the true measure of treatment success. Recovery of sexual function after RP was historically low, but nerve-sparing techniques have increased the possibility for many patients [1]. Current studies show that recovery rates range from 25–75% within a year of RP using nerve-sparing techniques, with younger and more sexually active patients having higher rates [2]. Despite nerve-sparing techniques, ED remains an inevitable complication of RP for prostate cancer. Researchers are investigating factors influencing sexual function recovery and the pathophysiology of iatrogenic ED, including nerve injury and neurapraxia [3]. Therapeutic approaches that have been studied include pluripotent stem cells and human amnion–chorion membrane allografts enriched with cytokines [4,5].

Mesenchymal stem cells (MSCs) possess long-term self-renewal capacity and can differentiate into different cell types under specific conditions [6,7]. In various physiological and pathological conditions, MSCs can aid in maintaining homeostasis by differentiating in multiple directions [7]. New MSC-based therapeutic strategies have demonstrated promising results in clinical practice, including the treatment of nerve damage, inflammation, and transplantation [8]. The MSC therapy for ED is still being studied in clinical trials [9]. Many studies have revealed positive outcomes of MSC therapy on ED [10]. Although MSC therapy has demonstrated positive outcomes for ED, the exact impact of MSCs on this condition, and the underlying mechanisms, remain incompletely understood [11,12]. Proper evaluation of the engraft of MSCs in the injection site and delivery methods have not been done yet [13,14,15].

The cell-derived matrix (CDM) plays a role in neural differentiation, influencing cell behavior and differentiation through physical and biochemical cues [16]. The composition and mechanical properties of the CDM can provide physical and biochemical cues that influence neural progenitor cell behavior and differentiation, including cell adhesion, migration, proliferation, and signaling pathways that control gene expression and cell death [17]. It can be used to create supportive scaffolds in tissue engineering, mimicking the natural microenvironment of developing neurons [18]. While CDM is easy to generate in vitro, its fragility and manipulation difficulties make it less practical for in vivo applications. Combining CDM with a mechanical platform for MSC engraftment may enhance its effects [19].

To investigate enhanced delivery of MSCs, we conducted a study using a CDM platform. This platform stamps CDM onto a mechanically-compliant polyvinyl alcohol (PVA)/polyethylene glycol (PEG) hydrogel (Figure 1A) with a human fibroblast-derived matrix (hFDM), stabilizing the structure and promoting nerve regeneration [20,21]. In this study, we evaluated the effects of a biomechanical CDM patch on hBMSCs in a BCNI rat model.

## 2. Materials and Methods

### 2.1. Preparation of Human Fibroblast-Derived ECM (hFDM) Patch

The fabrication of patches composed of hFDM and PVA was based on previously reported papers [21,22]. Human fibroblast (WI-38, CCL-75) cells were obtained from the American Type Culture Collection (Manassas, VA). Figure 1A shows a schematic of hFDM combined with PVA (upper) and a successful stamping image (lower). The state of the hFDM was assessed after each cycle using a light microscope (Axio Vert.A1; Carl Zeiss, Oberkochen, Germany) (Figure 1B).

### 2.2. Cell Culture and Differentiation

The Catholic Institute of Cell Therapy (Seoul, Republic of Korea) provided the Human bone marrow-derived mesenchymal stem cells (hBMSCs, Catholic MASTER Cells) that were used in this study. The hBMSCs were cultured in DMEM supplemented with 20% fetal bovine serum and 1% penicillin-streptomycin, in a 5% CO_2_ atmosphere at 37 °C. To induce neuronal differentiation, the cells were grown in a neuronal induction medium made up of neurobasal media, FBS (2%), 50 μM forskolin, epidermal growth factor, basic fibroblast growth factor, laminin, brain-derived neurotrophic factor, and penicillin-streptomycin. The differentiation medium was exchanged every 2 days up to 21 days.

### 2.3. Cell Adhesion and Proliferation on FN and hFDM Patch

PVA/hFDM hydrogel membranes with ECM facing upwards were placed in 12-well plates, and hBMSCs (2 × 10^4^) were seeded onto each substrate. Cell proliferation and viability were assessed using a Cell Counting Kit-8 (CCK-8; Dojindo, Tokyo, Japan) assay and live/dead viability assay kit (Invitrogen Life Technologies; Thermo Fisher Scientific, Inc., Waltham, MA, USA), respectively, 7 days after seeding. To perform the CCK-8 assay, 100 µL aliquots were taken from each sample and added to a 96-well plate. The plate was then incubated with 10% CCK-8 solution at 37 °C for 2 h, and the absorbance of each well was measured at 450 nm.

### 2.4. Immunocytochemistry

After neuronal differentiation induction, cells were fixed with 4% paraformaldehyde for 20min, and then permeabilized in 0.5% Triton X-100 for 10 min. The cells were blocked by incubation with bovine serum albumin (5%, 1 h), and then incubated overnight with primary anti-neuron-specific β-tubulin III (diluted 1:200, ab78078; Abcam, Cambridge, UK). Samples were incubated for 2 h with a secondary antibody (Alexa 488-conjugated goat anti-mouse) following washing. Confocal microscopy (LSM 800w/Airyscan; Carl Zeiss Inc., Oberkochen, Germany) was utilized

### 2.5. Quantitative Reverse Transcription-PCR (qRT-PCR)

Total RNA was extracted from differentiated cells using TRIzol^®^ reagent (Invitrogen Life Technologies; Thermo Fisher Scientific, Waltham, MA, USA) following the manufacturer’s protocol. Total RNA (1 µg) was reverse transcribed to cDNA using the a PrimeScriptTM RT reagent Kit (Takara Bio, Shiga, Japan) and The TB Green™ Premix Ex Taq™ II (Takara Bio, Shiga, Japan) was used for PCR amplification. The qRT-PCR was conducted with the following primers: Nestin, 5′-CCAGTTCTGCTCCTCTCCAG-3′(forward) and 5′-GCCCACAGATTCCTCTTCTG-3′(reverse); β-tubulin III, sense 5′-ATGGGATGGGTGTTCCTACA-3′(forward) and 5′-GTCTTAGAGAGGGCGACGTG-3′ (reverse); GFAP, 5′-CAACCACCCTCTTCACCACT-3′(forward) and 5′-GATCTTCTGGGGTGGTCTCA-3′(reverse); Glyceraldehyde 3-phosphate dehydrogenase (GAPDH), 5′-CAAGAACCCCAAGGACAAGA-3′(forward) and 5′-GAATCCATCGGTCATGCTCT-3′(reverse). Relative gene expression levels were determined using the 2^−ΔΔCT^ method and normalized to GAPDH, which served as an internal control.

### 2.6. Animal Experiments

Sprague–Dawley rats (8 weeks old) were obtained from a commercial vendor (Orient Bio, Seoul, Republic of Korea). Rats were randomly divided into five groups (*n* = 5 per group): an age matched normal control (Normal), BCNI, hBMSC (5 × 10^5^), hBMSC (1 × 10^5^) seeded on the hFDM (hFDM/hBMSC (1 × 10^5^)), and hBMSC (5 × 10^5^) seeded on the hFDM (hFDM/hBMSC (5 × 10^5^)) group. Rats were given a combination of Zoletil 50 (15 mg/kg, Virbac Laboratories, Carros, France) and Xylazine Hydrochlorid (5 mg/kg, Bayer, Seoul, Republic of Korea) for anesthesia. The BCNI rat model was used, which involved making a inferior midline incision to access from bladder to prostate and compressing the bilateral cavernous nerves for 2 min with a hemostat. To track the hBMSCs, they were labeled with red fluorescent dye PKH-67 (PKH-67 fluorescent cell linker kits; Sigma-Aldrich, St. Louis, MO, USA). 200 µL of hBMSCs (5 × 10^5^ cells) were then injected around each injured cavernous nerve in the hBMSC groups. Figure 1C illustrates the surgical application of a 10 mm patch to the MPG.

### 2.7. Erectile Function Measurement

After a 4-week period, intracavernosal pressure (ICP)/mean arterial pressure (MAP) was measured to assess erectile function. To measure ICP, a 23-gauge butterfly needle containing heparin solution (250 U/mL) was inserted into the proximal corpus cavernosum and connected to a pressure transducer. Then, the MPG was stimulated with a bipolar stainless-steel electrical stimulator for 50 s (10 V, 2.4 mA, 3.5 ms pulse). During nerve electrostimulation, a force transducer was used to measure the maximal peak of ICP. The recorded data was acquired by a data acquisition system (Power Lab; AD Instruments, Dunedin, New Zealand).

The measurement of MAP was taken by inserting PE-50 tubing (BD Intramedic, Franklin Lakes, NJ, USA) into the carotid artery. Half of each penis was fixed in paraformaldehyde (4%) and embedded within paraffin wax, and the other half was preserved at −70 °C.

### 2.8. Immunohistochemistry

For in vivo cell tracking, paraffin-embedded cavernous nerve sections were immunostained with the following primary antibodies: cell nuclei (DAPI; Vector Laboratories, Burlingame, CA, USA) and anti-β-tubulin III. The penile tissue was sectioned to a thickness of 4 μm for staining with Masson trichrome for smooth muscle and collagen. The paraffin sections were used for immunostaining analysis, and the following primary antibodies were applied: anti-neuronal nitric oxide synthase (nNOS, diluted 1:100, ab76067; Abcam, Cambridge, UK), anti-β-tubulin III, anti-α-smooth muscle actin (α-SMA, diluted 1:250, ab5694; Abcam, Cambridge, UK), anti-endothelial nitric oxide synthase (eNOS, diluted 1:100, ab5589; Abcam), anti-CD31 (diluted 1:50, ab28364; Abcam), and DAPI. Using a microscope (Zeiss, LSM 510 Meta Confocal, Oberkochen, Germany), fluorescent images were captured, and the mean fluorescent intensity was analyzed using ZEN2012 software (Zeiss, Germany). An optical microscope (Olympus, BX50, Tokyo, Japan) was used to obtain digital images, and GraphPad Prism v5 software (GraphPad Prism Software, San Diego, CA, USA) was used to calculate the mean fluorescent intensity.

### 2.9. Measurement of Cyclic Guanosine Monophosphate (cGMP) Levels

Penile tissue (50 mg) was homogenized with HCl (0.1 M, 300 mL) and silica beads (BioSpec Products, Inc., Guelph, ON, Canada), and then centrifuged at 4 °C and 12,000× *g* for 10 min. The collected supernatant was used to measure cGMP levels with the cGMP Direct Immunoassay Kit (K372-100; BioVision, Edmonton, Alverta, Canada).

### 2.10. Statistical Analyses

The statistical analysis was performed using GraphPad Prism v5 and the results were expressed as mean ± SD. One-way ANOVA was used to compare the differences between groups, followed by Tukey’s post-hoc test. A *p*-value of less than 0.05 was considered statistically significant.

## 3. Results

### 3.1. Characterization of PVA/hFDM

Figure 1A illustrates the method for incorporating PVA hydrogel with hFDM. The hFDM was prepared using a previously reported method [21,22,23], and the presence of the ECM fibrillar structure confirmed through optical microscopy (Figure 1B). PVA solution was then added to the hFDM and freeze–thaw induced physical cross-linking of the hydrogel [20]. A membrane of PVA/hFDM was then carefully removed from the plastic substrate using forceps, ensuring a secure coupling of hFDM with the PVA hydrogel.

### 3.2. Cell Viability and Cell Adhesion of hBMSC Seeded on Patch

The biological properties of the hFDM/PVA patch were evaluated via cell viability and cell adhesion. The hBMSCs were seeded on FN or hFDM-coated PVA and grown for 7days, followed by Live and Dead staining. As a result, hBMSC cells cultured on hFDM exhibited a higher percentage of viable cells, and they spread quite well with spindle-like morphology, compared to the control FN (Figure 2A). The cell viability rate was analyzed using the CCK-8 at 3 and 7 days of growth. We found that cell growth improved on hFDM, when compared with the FN on both days 3 and 7 (Figure 2B). Furthermore, immunofluorescence staining showed that the ECM on the hFDM/PVA membrane showed favorable integration with hBMSCs. (green: Fibronectin, red: Phalloidin). These findings suggested that the interaction between ECM and cells was stronger in the hFDM group compared to the FN group (Figure 2C). Figure 2D demonstrates that the cytoskeleton (red color) of cells in the hFDM layered group exhibited enhanced development compared to the FN group, with a statistically significant increase in both fibronectin and phalloidin staining. These findings suggested a significant upregulation of cytoskeletal organization in the hFDM layered group.

### 3.3. Cell Differentiation of hBMSC Seeded on Patch

To assess the effect of the hFDM patch on hBMSC neural differentiation, cells were seeded on either FN or hFDM-coated PVA and cultured for one to three weeks. Neural differentiation was evaluated by immunocytochemistry and qRT–PCR. The fluorescence microscope image (Figure 3A) demonstrates that hBMSCs cultured on hFDM-coated PVA exhibited greater expression of the neuronal marker β-tubulin III, compared to those cultured on FN-coated PVA. Subsequently, mRNA expression level was confirmed using qRT–PCR to assess the effect of PVA on neural differentiation. Similar to the immunocytochemistry result, mRNA expression of β-tubulin III cultured on hFDM for 1 week was higher than on the FN (Figure 3B). Moreover, it increased significantly with seeding on hFDM for 3 weeks compared to FN. The GFAP (glial marker) expression was also higher on the hFDM than on the FN. On the other hand, the level of nestin (neural stem cell marker) decreased at 3 weeks compared to 1 week, and was significantly decreased when cultured on hFDM rather than on the FN. Our results demonstrated that the hFDM induced differentiation of neural stem cells.

### 3.4. The hBMSCs Seeded on Patches Improves Erectile Function

To investigate the potential of the patch to treat ED in the BCNI rat model, hBMSCs seeded on patches were attached to the injured cavernous nerve (Figure 4A). Representative ICP curves and ICP/MAP ratios for all groups are shown in Figure 4B,C. The ICP/MAP ratios of all hBMSC groups, including the hBMSC (5 × 10^5^), hFDM/hBMSC (1 × 10^5^) and hFDM/hBMSC (5 × 10^5^) groups, were significantly higher, when compared to the BCNI group. The ICP/MAP ratios in the hBMSC (5 × 10^5^) and hFDM/hBMS (1 × 10^5^) groups were similarly increased despite the lower cell seeding density of the hFDM/hBMSC (1 × 10^5^) group. Moreover, the ICP/MAP ratio was significantly increased in the hFDM/hBMSC (5 × 10^5^) group compared to the hBMSC-only group, when seeded at the same density. These results indicated that the patch could effectively improve erectile function compared to hBMSC cells alone.

To confirm hBMSC survival, we investigated PKH-67 labeled hBMSCs. PKH-67 labeled hBMSC (red) were localized around the cavernous nerve injury site in all hBMSC groups, including the hBMSC (5 × 10^5^), hFDM/hBMSC (1 × 10^5^) and hFDM/hBMSC (5 × 10^5^) groups (Figure 5).

### 3.5. hBMSCs Seeded in the Patch Increases Smooth Muscle Cells in the Corpus Cavernosum

Figure 6A,B shows the representative images of smooth muscle in the corpus cavernosum, obtained through Masson’s trichrome staining to evaluate its association with erectile dysfunction. Compared to the normal group, the BCNI group caused penile fibrosis in the corpus cavernosum, which was demonstrated by a decrease in smooth muscle content (red) and an increase in collagen content (blue). On the other hand, the hBMSC (5 × 10^5^) and hFDM/hBMSC (1 × 10^5^) groups showed an increase in the SMA/collagen ratio, whereas it was reduced in the BCNI group.

Surprisingly, the hFDM/hBMSC (5 × 10^5^) group exhibited significantly increased SMA/collagen ratios compared to the respective expression levels of the hBMSC (5 × 10^5^) and hFDM/hBMSC (1 × 10^5^) groups. Additionally, the level of α-SMA, a smooth muscle cell marker, was determined by immunohistochemical staining in the corpus cavernosum. Consistent with the previous results, expressions of α-SMA in the hBMSC (5 × 10^5^) and hFDM/hBMSC (1 × 10^5^) groups were higher than in the BCNI group, and α-SMA in the hFDM/hBMSC (5 × 10^5^) group was notably increased, compared to the hBMSC (5 × 10^5^) and hFDM/hBMSC (1 × 10^5^) groups (Figure 6C,D). These results meant that the patch improved erectile function by increasing smooth muscle cells.

### 3.6. hBMSCs Seeded in the Patch Restores Nitric Oxide (NO)/cGMP Signaling Pathway

The nNOS and eNOS are two factors that most produce NO. As shown in Figure 7A, nNOS expression (red) decreased after nerve injury when compared to the normal group. However, nNOS in the hBMSC (5 × 10^5^) and hFDM/hBMSC (1 × 10^5^) groups showed higher expressions than in the BCNI group, and the nNOS levels were significantly higher in the hFDM/hBMSC (5 × 10^5^) group than in the other two groups (Figure 7B). Next, we performed immunohistochemical staining for eNOS (red) and CD31 (green: Endothelial marker) to evaluate the eNOS expression and endothelial cells in the corpus cavernosum. The hBMSC (5 × 10^5^), hFDM/hBMSC (1 × 10^5^), and hFDM/hBMSC (5 × 10^5^) groups showed increased expressions of eNOS, compared to the BCNI group, as depicted in Figure 7C. Furthermore, cGMP levels in the corpus cavernosum, as measured using the cGMP Direct Immunoassay Kit, were significantly higher in the hBMSC (5 × 10^5^), hFDM/hBMSC (1 × 10^5^), and hFDM/hBMSC (5 × 10^5^) groups, compared to the BCBI group (Figure 7D). These results indicated that the patch enhanced the NO/cGMP pathway, thereby improving erectile function.

## 4. Discussion

The clinical challenges of regenerating and repairing the cavernous nerve after RP have not yet been successfully addressed. Yiou et al. [10] conducted a clinical trial evaluating the efficacy of MSCs obtained from the iliac crest and injected in the cavernosum of the penis, but not directly into the cavernous nerve injury site. This study found that MSC injection improved erectile function after one-year follow-up, providing a promising outcome. According to Liang et al. [24], one of the main challenges in using MSCs for treatment is that the transplanted cells may not effectively reach, and function at, the targeted area. Many other studies [10,25,26] have used cavernosum injection as a method of delivering MSCs, but this approach is associated with the potential risk of MSCs migrating into the circulatory system and failing to accumulate in adequate concentrations at the targeted injury site, resulting in suboptimal therapeutic effects [27,28]. Therefore, it is more advantageous to directly deliver MSCs to the cavernous nerve at the actual injury site so the delivered MSCs can differentiate into neural cells.

In this study, we assessed the effects of MSCs in vitro and in vivo using a patch layered on a PVA hydrogel surface. The patch, which combines MSCs and CDM, provides physical, topographical, and chemical signals for cell attachment and proliferation through protein synthesis. The matrix-derived collagen, fibronectin, laminin, and other biomaterial components create a chemical microenvironment that favors the differentiation of specific cells [29]. We combined MSCs and CDM with a patch and attached this to the site of cavernosal nerve injury. While higher cell seeding densities were typically used in previous studies [25,30,31], we intentionally investigated the potential of a patch with lower cell density to enhance erectile function. Our hypothesis was that the patch could improve cell retention at the site of nerve injury, even with reduced cell density. The results supported this hypothesis, as the hFDM/hBMSC (5 × 10^5^) group exhibited significantly higher ICP/MAP ratios than the hBMSC group, despite the equal cell density. This suggests that the patch, combined with hBMSC, can achieve comparable improvements in erectile function. Although specific analyses for VEGF, TGF-beta, and Caspase-3 were not performed in this study, our results confirmed the enhancement of erectile function through the observed improvements in neuronal differentiation, as shown in the in vitro experiments (Figure 3) and in vivo experiments (Figure 5). These findings suggest that the patch-mediated hBMSCs application may influence factors related to fibrosis, apoptosis, neurogenesis and angiogenesis.

However, the European Society for Sexual Medicine stated that stem cells may accelerate the recovery process, instead of increasing the overall recovery rate [32]. Many studies evaluated the recovery of erectile function just 4–12 weeks after nerve injury, disregarding the natural improvement in erectile function, which can take several months after BNCI. Moreover, most studies used young rats without comorbidities, whereas prostate cancer patients are generally older and may have other comorbidities that could affect recovery rates after RP.

A recent study by Castiglione et al. [33] investigated the long-term effects of BNCI rats with type 1 diabetes and found that, unlike healthy rats, diabetic rats did not show natural recovery of erectile function at 4 months after BNCI. Zhang et al. [28] suggested that stem cell injection is effective in treating diabetes-induced ED in a rat model. This study revealed that adipose-derived stem cells have therapeutic potential for improving ED by differentiating into smooth muscle-like cells.

The effect of neuronal differentiation of MSCs transplanted into the patch for nerve injury repair remains unclear, but the use of a mechanically stable patch has shown promise in enhancing MSC biological function and improving erectile function. Further research is needed to validate these findings.

## 5. Conclusions

In conclusion, the novel patch was made by combining hFDM with PVA to create a mechanically robust and biocompatible patch. The synergistic effects observed highlight the importance of both the mechanical support, provided by the patch, and the therapeutic potential of hBMSC in improving angiogenesis, and smooth muscle and nitrergic nerve regeneration. The PVA/hFDM patch shows potential for nerve regeneration as a promising approach for post prostatectomy ED.

## Figures and Tables

**Figure 1 bioengineering-10-00635-f001:**
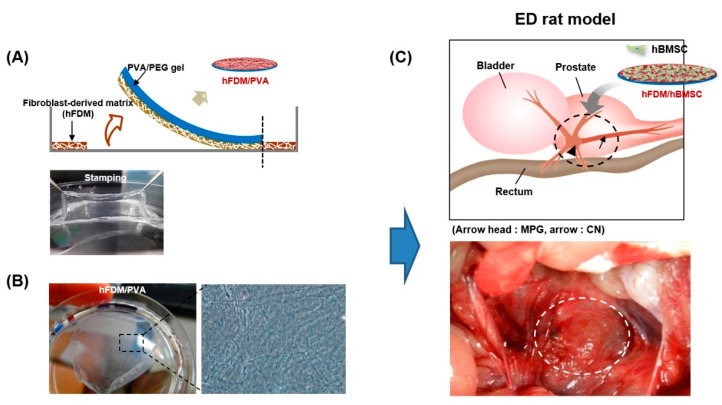
Fabrication of hFDM-loaded PVA/PEG gel. (**A**) Schematic illustration in fabricating hFDM-loaded PVA/PEG gel. (**B**) After physical stimulation, the nanofibrous structure on the surface of the hFDM/PVA membrane was confirmed through phase contrast images. Stamping image and microscopic image of hFDM/PVA gel. (**C**) Schematic diagram showing application of hBMSC attached to hFDM/PVA gel on injured CN. hFDM, human fibroblast-derived matrix; PVA, polyvinyl alcohol; PEG, polyethylene glycol; hBMSC, human bone marrow-derived mesenchymal stem cells; MPG, major pelvic ganglion; CN, cavernous nerve.

**Figure 2 bioengineering-10-00635-f002:**
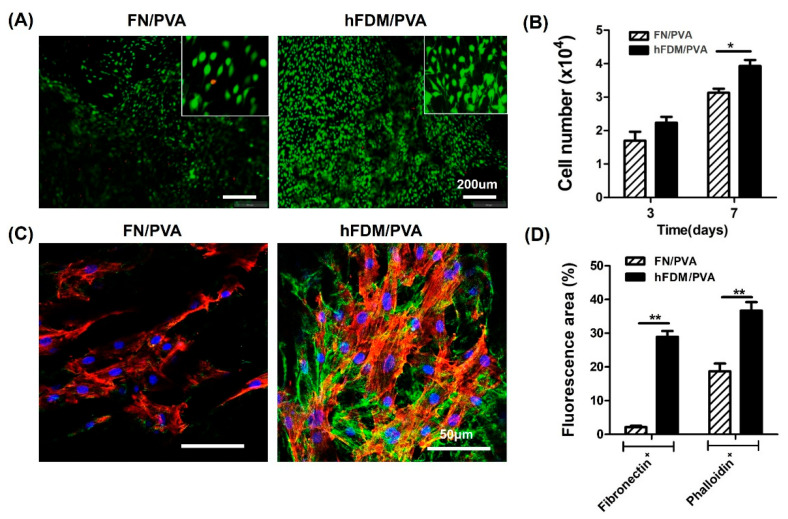
Characterization of hFDM/PVA patch. (**A**) Cell viability of hBMSC after 7 days culture on each substrate was evaluated via live/dead assay. Live cells were stained green and dead cells were stained blue. Scale bars are 200 μm. The inserts are enlarged-images enlarged with the built-in enlarging feature in the software of the microscope. (**B**) Cell proliferation was determined by counting the number of cells after 3 days or 7 days of culturing on each substrate. Data is means ± SD. * *p* < 0.05. (**C**) Immunofluorescence of fibronectin (green) and phalloidin (red) in the FN/PVA and hFDM/PVA. hBMSCs were cultured on each substrate for 3 days, and cell adhesion detected by immunofluorescence staining. Nuclei were counterstained with 4′,6-diamidino-2-phenylindole (DAPI) (blue). Scale bars are 50 μm. (**D**) Quantification of fibronectin and phalloidin staining in the hFDM layered group compared to the FN group. Data are means ± SD. ** *p* < 0.05.

**Figure 3 bioengineering-10-00635-f003:**
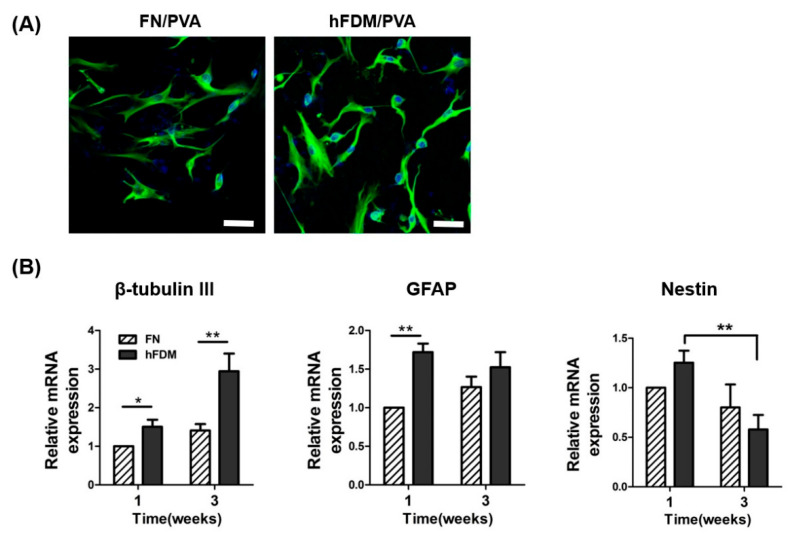
Neuronal differentiation of hBMSCs seeded on hFDM/PVA patch. (**A**) The neural differentiation of hBMSCs was induced by growing cells of each substrate in neural induction medium for 1 week. Mature neurons were visualized with β-tubulin III (green), and the nuclei were visualized with 4′,6-diamidino-2-phenylindole (DAPI) (blue). Scale bar is 50 μm. (**B**) Expression of mRNA for nestin, GFAP, and β-tubulin III were quantified by qRT-PCR. GAPDH mRNA was used as an internal control to normalize the data. Data are means ± SD. * *p* < 0.05, ** *p* < 0.05.

**Figure 4 bioengineering-10-00635-f004:**
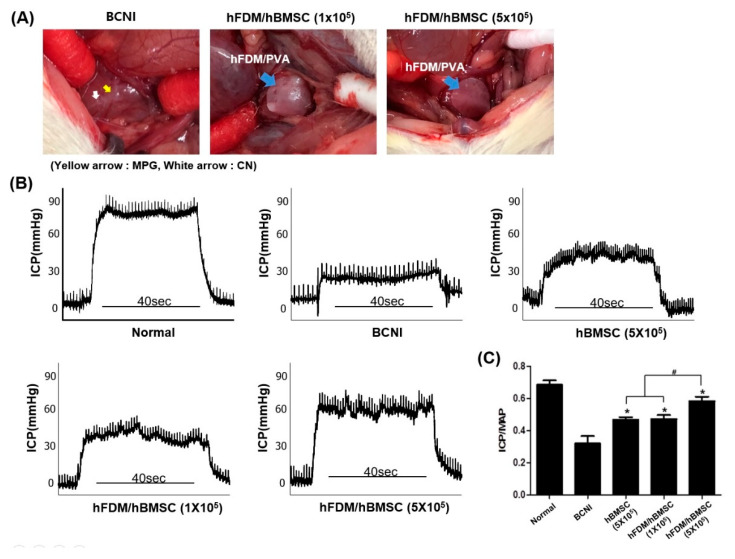
Erectile function assessment of hFDM/PVA patch. (**A**) Representative image of the hFDM/hBMSC attachment to the injured CN in the BCNI model. (**B**) Representative ICP curves for the normal group, BCNI group, hBMSC (5 × 10^5^) group, hFDM/hBMSC (1 × 10^5^) group and hFDM/hBMSC (5 × 10^5^) group. ICP curve recorded in response to electrostimulation of the cavernous nerve for 40 s. (**C**) The bar graph shows the ratio of ICP to MAP. Each bar represents mean ± SD (*n* = 5 animals per group). * *p* < 0.05 compared with BCNI, # *p* < 0.05 compared with hBMSC (5 × 10^5^) or hFDM/hBMSC (1 × 10^5^). BCNI, bilateral cavernous nerve injury; ICP, intracavernosal pressure; MAP, mean arterial pressure.

**Figure 5 bioengineering-10-00635-f005:**
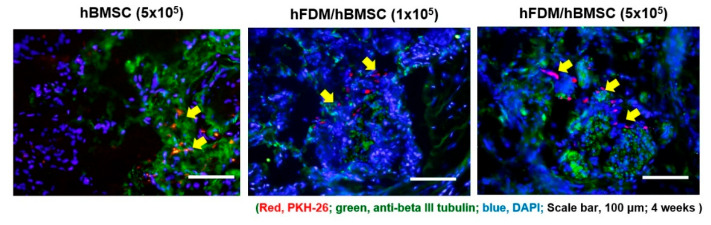
In vivo tracking of PKH67-labeled hBMSCs seeded on hFDA/PVA patch in BCNI model. Cells were immunostained using β-tubulin III (green) and PKH-67 (red) 4 weeks after surgery, and the nuclei were visualized with 4′,6-diamidino-2-phenylindole (DAPI) (blue). PKH67-labeled hBMSCs are indicated by arrows. Scale bar is 100 μm.

**Figure 6 bioengineering-10-00635-f006:**
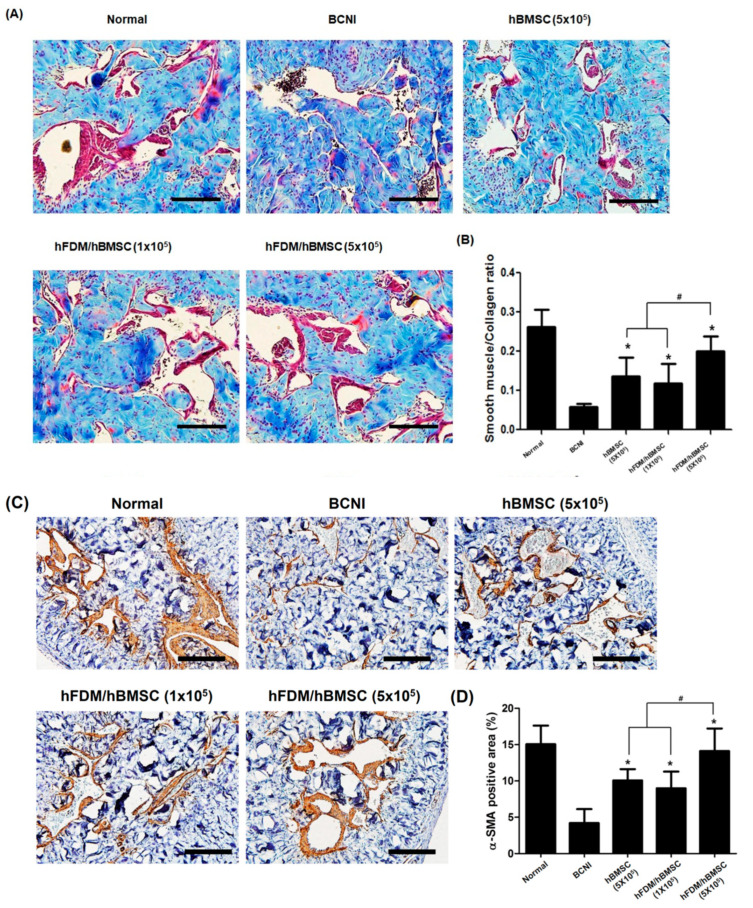
Change of smooth muscle content in corpus cavernosum tissue. (**A**) Representative images showing Masson trichrome staining in all five groups. Smooth muscle and collagen were stained in red and blue, respectively. (**B**) The bar graph shows the smooth muscle/collagen ratio (mean ± SD) (**C**) Representative results of immunohistochemistry for alpha smooth muscle actin (α-SMA) in all five groups. α-SMA was stained brown, and nuclei were counterstained with hematoxylin (blue). (**D**) The bar graph represents a quantitative assessment of α-SMA expression (mean ± SD). Magnification, ×200. * *p* < 0.05 compared with BCNI, # *p* < 0.05 compared with hBMSC (5 × 10^5^) or hFDM/hBMSC (1 × 10^5^).

**Figure 7 bioengineering-10-00635-f007:**
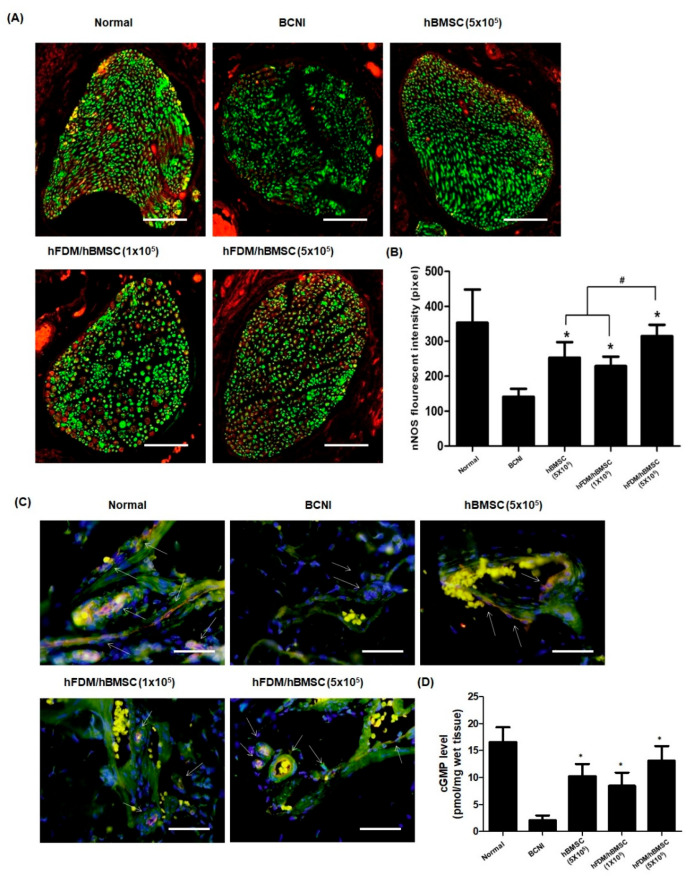
Expression of nNOS and eNOS, and level of cGMP in corpus cavernosum. (**A**) Representative images of immunohistochemical staining in the dorsal penile nerve. nNOS was stained red and neuron-specific β-tubulin III was stained green (magnification, ×400). (**B**) Quantification of nNOS expression, expressed as the fluorescence intensity of the dorsal penile nerve in each group (mean ± SEM). (**C**) Representative images of immunohistochemical staining in the corpus cavernosum. The eNOS was stained red and endothelial-specific CD31 was stained green (magnification, ×400). (**D**) The cGMP levels in the corpus cavernosum were measured with a cGMP Direct Immunoassay Kit. Each bar represents mean ± standard error. * *p* < 0.05 compared with BCNI, # *p* < 0.05 compared with hBMSC (5 × 10^5^) or hFDM/hBMSC (1 × 10^5^). The nNOS, neuronal nitric oxide synthase; eNOS, endothelial nitric oxide synthase; cGMP, cyclic guanosine monophosphate.

## Data Availability

Not applicable.

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
