# Peer review of "Erectile Dysfunction Treatment Using Stem Cell Delivery Patch in a Cavernous Nerve Injury Rat Model"

_bioengineering, 2023, doi:10.3390/bioengineering10060635_

Round 1

Reviewer 1 Report

It is a well planned study. Idea is innovative as authors have used stem cell patch. However, how far it would be practical to use in clinical condition is big question mark as it require surgery. Although, this paper is very nicely written, but I have few important comments to make:

Authors have studied structural changes like smooth muscle/collagen ratio and smooth muscle cell markers before and after application of stem cell patch. However, it would have been better if authors would have analyze TGF-beta and Caspase-3 in order to prove that it indeed reduces fibrosis and apoptosis. 

The rat model was developed by compressing cavernous nerve by hemostat. There is a possibility to develop ischemic and hypoxic state of corpus cavernosum which may increase reactive oxygen species and that may cause apoptosis. The improvement in penile ischemia is a important factor in ED progress and hence level of Caspase-3 is important in this study. 

Authors should write limitation of this study as you generally get high level of heterogeneity in ICP/MAP outcome. 

Vascular endothelium and VEGF are the basic pathway for stem cell therapy for cavernosum nerve injury (CNI). It is therefore important to study vascular endothelium and VEGF for mechanism in the pathogenesis and recovery in ED. Several papers reported that endothelial repair is a main reason for ED recovery in CNI-ED model beside nNOS. In fact, VEGF promote neural repair and hence cannot miss to analyze. 

Paper need minor English revision.  

Minor revision required

Author Response

1>     It is a well planned study. Idea is innovative as authors have used stem cell patch. However, how far it would be practical to use in clinical condition is big question mark as it require surgery. Although, this paper is very nicely written, but I have few important comments to make:

A>    We appreciate the reviewer's concern about the practicality of the stem cell patch approach in clinical conditions. It does not require additional surgery, as it can be applied to the nerve injury site during the radical prostatectomy procedure. Similar approaches, such as the use of amniotic membrane patches, are being studied and implemented shortly after surgery [1]. These patches provide a source of neurotrophic factors and cytokines.
References
1.    Patel, V.R., et al., Dehydrated Human Amnion/Chorion Membrane Allograft Nerve Wrap Around the Prostatic Neurovascular Bundle Accelerates Early Return to Continence and Potency Following Robot-assisted Radical Prostatectomy: Propensity Score-matched Analysis. Eur Urol, 2015. 67(6): p. 977-980.

2>    Authors have studied structural changes like smooth muscle/collagen ratio and smooth muscle cell markers before and after application of stem cell patch. However, it would have been better if authors would have analyze TGF-beta and Caspase-3 in order to prove that it indeed reduces fibrosis and apoptosis. The rat model was developed by compressing cavernous nerve by hemostat. There is a possibility to develop ischemic and hypoxic state of corpus cavernosum which may increase reactive oxygen species and that may cause apoptosis. The improvement in penile ischemia is a important factor in ED progress and hence level of Caspase-3 is important in this study. 

A>    Although we did not directly measure TGF-beta and Caspase-3 in this study, the effects of hBMSCs on fibrosis and apoptosis have been well-documented in previous research [2, 3]. TGF-beta is a key factor involved in fibrosis, and Caspase-3 is associated with apoptosis, which are important processes in tissue damage caused by hypoxia and ischemia. However, it is important to note that fibrosis ultimately occurs due to excessive collagen accumulation. 
Our focus was on investigating the effects of human bone marrow-derived mesenchymal stem cells (hBMSCs) in preventing the deterioration of corporeal smooth muscle and collagen deposition. These hBMSCs have demonstrated antiapoptotic and antifibrotic effects in previous studies [2, 3]. Additionally, we observed the upregulation of neuronal nitric oxide synthase (nNOS), which supports the positive impact of hBMSCs on neural differentiation and function. Furthermore, we evaluated the levels of cyclic guanosine monophosphate (cGMP), a key molecule involved in the regulation of penile erection.
Therefore, our study aimed to evaluate the preservation of smooth muscle and collagen deposition, as well as the promotion of neuronal differentiation through the recovery of nNOS and cGMP. 
References
2.    Fall, P.A., et al., Apoptosis and effects of intracavernous bone marrow cell injection in a rat model of postprostatectomy erectile dysfunction. Eur Urol, 2009. 56(4): p. 716-25.
3.    Kim, S.G., et al., Therapeutic Effect of Human Mesenchymal Stem Cell-Conditioned Medium on Erectile Dysfunction. World J Mens Health, 2022. 40(4): p. 653-662.

3>    Authors should write limitation of this study as you generally get high level of heterogeneity in ICP/MAP outcome. 

A>    We have made revisions to the discussion section regarding the effects of cell density and patch application on erectile function improvement and the hypothesis for heterogeneity observed in the ICP/MAP results.  (line 350-362)
B>    Revised manuscript
“While higher cell seeding densities are typically used in previous studies, we intentionally investigated the potential of a patch with lower cell density to enhance erectile function. Our hypothesis was that the patch could improve cell retention at the site of nerve injury, even with reduced cell density. The results supported this hypothesis, as the hFDM/hBMSC (5×105) group exhibited a significantly higher ICP/MAP ratio than the hBMSC group, despite the equal cell density. This suggests that the patch, combined with hBMSC, can achieve comparable improvements in erectile function. Although specific analyses for VEGF, TGF-beta, and Caspase-3 were not performed in this study, our results confirmed the enhancement of erectile function through the observed improvements in neuronal differentiation, as shown in the in vitro experiments (Fig. 3) and in vivo experiments (Fig. 5). These findings suggest that the patch-mediated hBMSCs application may have influenced factors related to fibrosis, apoptosis, neurogenesis and angiogenesis.”

4>    Vascular endothelium and VEGF are the basic pathway for stem cell therapy for cavernosum nerve injury (CNI). It is therefore important to study vascular endothelium and VEGF for mechanism in the pathogenesis and recovery in ED. Several papers reported that endothelial repair is a main reason for ED recovery in CNI-ED model beside nNOS. In fact, VEGF promote neural repair and hence cannot miss to analyze. 

A>     We appreciate the suggestion and agree with the importance of evaluating VEGF analysis, as it plays a crucial role in stem cell therapy for cavernous nerve injury. VEGF is known to have therapeutic effects in promoting tissue repair and angiogenesis. In our study, while focusing on the role of stem cells, we also investigated the specific effect of the patch in enhancing cell retention at the injury site. By maximizing the local concentration of stem cells, the patch facilitates their therapeutic effects and contributes to tissue regeneration. It is well known that MSCs have the capability to secrete VEGF. Therefore, we can speculate that the hBMSCs, which were sufficiently retained at the injury site, could have expressed VEGF. We will persist in our research in this field and, in future studies, we plan to include the analysis of TGF-beta, Caspase-3, TUNEL and VEGF, which are related to fibrosis and apoptosis, key factors in the recent therapeutic strategies beyond the smooth muscle cells in ED with CNI models.

5>    Paper need minor English revision.  

A>    I have conducted some minor English revision and completed the correction of typographical errors and spacing.

Reviewer 2 Report

This article mainly describes the Erectile Dysfunction that hFDM/PVA + hBMSCs patches can treat in rats. The suggestions are as follows:

1.This article describes that hFDM/PVA + hBMSCs patches can improve the erectile function of rats in various aspects, but the specific mechanism is not explained, which is not enough for a complete article. Please add a related mechanism to connect all the data together.

2.Please perform quantitative analysis on Figure 2-C

3.Please add C in Figure 6.

4.In Figure 7-C, the text stated that eNOS was stained red and endothelial-specific CD31 was stained green. I think the red signal is more like the expression of CD31. please confirm.

Author Response

This article mainly describes the Erectile Dysfunction that hFDM/PVA + hBMSCs patches can treat in rats. The suggestions are as follows

1.    This article describes that hFDM/PVA + hBMSCs patches can improve the erectile function of rats in various aspects, but the specific mechanism is not explained, which is not enough for a complete article. Please add a related mechanism to connect all the data together.

A> We agree that providing a detailed mechanism underlying the observed improvement in erectile function is essential. In the revised manuscript, we will include a comprehensive explanation of the related mechanism. (line 352-361)

B> Revised manuscript : While higher cell seeding densities are typically used in previous studies [25, 30, 31], we intentionally investigated the potential of a patch with lower cell density to en-hance erectile function. Our hypothesis was that the patch could improve cell retention at the site of nerve injury, even with reduced cell density. The results supported this hypothesis, as the hFDM/hBMSC (5×105) group exhibited a significantly higher ICP/MAP ratio than the hBMSC group, despite the equal cell density. This suggests that the patch, combined with hBMSC, can achieve comparable improvements in erec-tile function. Although specific analyses for VEGF, TGF-beta, and Caspase-3 were not performed in this study, our results confirmed the enhancement of erectile function through the observed improvements in neuronal differentiation, as shown in the in vitro experiments (Fig. 3) and in vivo experiments (Fig. 5). These findings suggest that the patch-mediated hBMSCs application may have influenced factors related to fibrosis, apoptosis, neurogenesis and angiogenesis.

   2.    Please perform quantitative analysis on Figure 2-C

A> We acknowledge the suggestion for quantitative analysis of Figure 2-C. In the modified Figure 2-D, we quantified the positive areas of anti-fibronectin and anti-phalloidin staining separately and calculated the ratio of the respective positive areas, since the yellow color co-localization was not clearly discernible, making quantification challenging. Our results demonstrate that the cytoskeleton (red color) of cells in the hFDM layered group exhibited enhanced development compared to the FN group, with a statistically significant increase in both fibronectin and phalloidin staining. These findings suggest a significant upregulation of cytoskeletal organization in the hFDM layered group.

3.    Please add C in Figure 6.

A>     We appreciate your observation regarding Figure 6-C. We will include the missing component (C) in Figure 6.

4.    In Figure 7-C, the text stated that eNOS was stained red and endothelial-specific CD31 was stained green. I think the red signal is more like the expression of CD31. Please confirm.

A>     Regarding Figure 7-C, we acknowledge the potential confusion in the text description. We confirm that the red signal in Figure 7-C represents the staining of eNOS, while the green signal corresponds to the endothelial-specific CD31 staining. We will revise the manuscript to clarify this point and provide a more accurate description.

Round 2

Reviewer 2 Report

Thanks for the point to point reply.